Patient-centered care interventions to reduce the inappropriate prescription and use of benzodiazepines and z-drugs: a systematic review

Mokhar Aliaksandra 1 a.mokhar@uke.de
Topp Janine 2
http://orcid.org/0000-0001-7443-9890 Härter Martin 1
Schulz Holger 1
Kuhn Silke 3
Verthein Uwe 3
http://orcid.org/0000-0002-5729-7357 Dirmaier Jörg 1
1 Department of Medical Psychology, University Medical Center Hamburg-Eppendorf , Hamburg , Germany
2 Department of Health Services Research in Dermatology and Nursing, University Medical Center Hamburg-Eppendorf , Hamburg , Germany
3 Department of Psychiatry and Psychotherapy, Center for Interdisciplinary Addiction Research, University Medical Center Hamburg-Eppendorf , Hamburg , Germany
Tulkens Paul
Electronic publication date: 2018 Oct 8
Publication date: 2018
Volume: 6
Electronic Location ID: e5535
Received 2018 Jan 17; Accepted 2018 Aug 8
Copyright: © 2018 Mokhar et al.
Copyright year: 2018
Copyright holder: Mokhar et al.
License: This is an open access article distributed under the terms of the Creative Commons Attribution License, which permits unrestricted use, distribution, reproduction and adaptation in any medium and for any purpose provided that it is properly attributed. For attribution, the original author(s), title, publication source (PeerJ) and either DOI or URL of the article must be cited.
License URL: https://creativecommons.org/licenses/by/4.0/

Keywords: Benzodiazepines, z-drugs, Inappropriate prescription, Long-term use, Older population, Health care professionals, Patient-centered care

Funding: The authors received no funding for this work.

==============================
Background

Benzodiazepines (BZDs) and z-drugs are effective drugs, but they are prescribed excessively worldwide. International guidelines recommend a maximum treatment duration of 4 weeks. Although these drugs are effective in the short-term, long-term BZD therapy is associated with considerable adverse effects, the development of tolerance and, finally, addiction. However, there are different interventions in terms of patient-centered care that aim to reduce the use of BZDs and z-drugs as well as assist health care professionals (HCPs) in preventing the inappropriate prescription of BZDs.

Aim

The aim of this systematic review was to identify interventions that promote patient-centered treatments for inappropriate BZD and z-drug use and to analyze their effectiveness in reducing the inappropriate use of these drugs.

Methods

To identify relevant studies, the PubMed, EMBASE, PsycINFO, Psyndex, and Cochrane Library databases were searched. Studies with controlled designs focusing on adult patients were included. Trials with chronically or mentally ill patients were excluded if long-term BZD and z-drug use was indicated. Study extraction was performed based on the Cochrane Form for study extraction. To assess the quality of the studies, we used a tool based on the Cochrane Collaboration’s tool for assessing the risk of bias in randomized trials.

Results

We identified 7,068 studies and selected 20 for systematic review. Nine interventions focused on patients, nine on HCPs, and two on both patients and HCPs. Intervention types ranged from simple to multifaceted. Patient-centered interventions that provided patient information effectively increased the appropriate use of BZDs. The educational approaches for HCPs that aimed to achieve appropriate prescription reported inconsistent results. The methods that combined informing patients and HCPs led to a significant reduction in BZD use.

Conclusions

This is the first review of studies focused on patient-centered approaches to reducing the inappropriate prescription and use of BZDs and z-drugs. The patient-centered dimension of patient information was responsible for a decrease in BZD and z-drug consumption. Further, in some studies, the patient-centered dimensions responsible for reducing the prescription and use of BZDs and z-drugs were the clinician’s essential characteristics and clinician-patient communication.

Introduction

Benzodiazepines (BZDs) and z-drugs (BZD derivatives, e.g., zolpidem and zopiclone) are among the most commonly used anxiolytics and hypnotics worldwide (Fassaert et al., 2007; Rogers et al., 2007). While BZD and z-drugs have been demonstrated to be effective in short-term use (Canadian Agency for Drugs and Technologies in Health, 2014), their intake is associated with serious adverse effects, including increased risk of cognitive impairments (Barker et al., 2004; McAndrews et al., 2003; Paterniti, Dufouil & Alperovitch, 2002) as well as stumbling and falling, which may result in hip fractures (Takkouche et al., 2007; Zint et al., 2010) as withdrawal symptoms (Rickels et al., 1990). The main serious problem associated with long-term use is the development of tolerance and dependence (Ashton, 2005; Voyer et al., 2009; Zint et al., 2010). The risks and adverse effects of BZDs are of particular relevance to older people. Therefore, the Beers Criteria Update Expert Panel for potentially inappropriate medication use recommends avoiding the prescription of BZDs to patients over the age of 65 years, regardless of their primary disease or symptoms (American Geriatrics Society Beers Criteria Update Expert Panel, 2012). Although guidelines and expert consensus confirm the risks associated with the long-term use of BZD, these drugs are still prescribed frequently (Fassaert et al., 2007; Rogers et al., 2007). Thus, despite increasing awareness of the associated risks, the prevalence of inappropriate use has not declined (Cunningham, Hanley & Morgan, 2010; Huerta et al., 2015).

“Inappropriate” BZD use is defined as BZD use that is associated with a significantly higher risk of adverse effects than treatment with an alternative evidence-based intervention that is equally, if not more, effective (Beers & Ouslander, 1989; Opondo et al., 2012). Different motives have been given for the inappropriate use of BZDs. Patients report that they lack information on alternative pharmacological and nonpharmacological treatment options, the discontinuation of BZDs, and the potentially hazardous effects of inappropriate BZD use (Beers & Ouslander 1989; Fang et al. 2009). Furthermore, regarding the patients perspective, they are often unwilling to discontinue BZD use, as possible physiological and psychological dependencies might be present (Fang et al., 2009; Tannenbaum et al., 2014). Different reasons for the inappropriate prescription of BZDs have been assessed (Anthierens et al., 2007b; Opondo et al., 2012; Voyer et al., 2009). These reasons include lack of knowledge of possible evidence-based alternative treatment options, nonspecific knowledge about BZDs among physicians and other specialists, especially in geriatric care, a lack of clarity about how to appropriately prescribe the drug and difficulties applying medication guidelines to clinical practice (Ashton, 2005; Opondo et al., 2012). Although physicians report being cautious about initiating BZD treatments, the psychosocial problems of patients are often severe, and the knowledge of how to handle these severe problems using alternative strategies is often limited (Anthierens et al., 2007a; Parr et al., 2006). Given the variety of severe risks and adverse effects, including possible dependency, the high prevalence of BZD use in older people in general and the high number of long-term users in particular, interventions that address this issue need to be identified (Gould et al., 2014; Oude Voshaar et al., 2006; Smith & Tett, 2010). To address this need, numerous studies have focused on the difficulties in physician-patient communication and patient information involved in the inappropriate use and prescription of BZDs. These studies have investigated specific interventions that are designed to educate patients, provide patient information material, improve physician-patient communication, or build a relationship between patients and physicians (Gould et al., 2014; Mugunthan, McGuire & Glasziou, 2011). These interventions can be considered to fall under the umbrella term patient-centeredness (Scholl et al., 2014; Zill et al., 2015). Patient-centered care is a comprehensive care concept (Bardes, 2012). Various definitions have tried to encompass the complexity of this idea (Scholl et al. 2014; Zill et al. 2015; Mead & Bower, 2000). Recently, Scholl et al. (2014) merged existing definitions and developed a comprehensive model of patient-centeredness. These researchers defined 15 dimensions of patient-centeredness and, according to expert consensus, isolated the five most relevant dimensions (Scholl et al., 2014). In addition to being treated as a unique individual, the patient’s involvement in his or her own care, patient empowerment, patient information, and clinician-patient communication were rated as the most relevant aspects (Zill et al., 2015). The latter dimensions are mainly understood to be the activities of patient-centered care, which has become an international demand for high-quality medicine (Mead & Bower, 2000; Phelan, Stradins & Morrison, 2001).

An increased emphasis on patient-centeredness could address the causes of inappropriate BZD use and decrease its prevalence by focusing on patients’ values. Patients’ beliefs, preferences, and information need to play a greater role in the care process. Putting the individual patient rather than his or her disease at the center of the treatment plan has increasingly been advocated, and numerous medical experts recommend the implementation of this strategy in routine care (Committee on Quality of Health Care in America IoMI, 2001). Research in various sectors of health care attests to improved care processes as a result of patient-centered approaches. Patients have reported that such approaches restored their satisfaction and self-management abilities and significantly improved their quality of life (Rathert, Wyrwich & Boren, 2012).

Research of the physician’s perspective describes the need for professional expertise, specific communication skills, and the ability to inform patients based on the evidence-based knowledge presented in guidelines and expert consensuses for clinical practice. Some studies have found that good physician-patient communication is associated with important patient health outcomes (Mercer et al., 2008; Zolnierek & Dimatteo, 2009). In addition to dimensions regarding physicians’ abilities, there are communication factors related to patient-centered activities where physicians provide information and better educate patients by sharing specific information and using informational resources and tools (Scholl et al., 2014). Furthermore, recent research indicates that interventions that promote patient-centered care have a positive influence on patient-related outcomes (Dwamena et al., 2012; Mead & Bower, 2002).

The high prevalence of inappropriate BZD use and the possible reasons for this use combined with the knowledge of the general benefits of a patient-centered approach in health care highlight the need to consider a patient-centered approach for patients using BZDs. By focusing on the five most important aspects of patient-centered care, this systematic review aimed to identify patient-centered interventions for reducing the inappropriate prescription and use of BZDs and z-drugs.

Methods

This systematic review was registered with the International Prospective Register of Systematic Reviews (PROSPERO): CRD42014015616. The reporting guidelines used for this review were based on the Preferred Reporting Items for Systematic Reviews and Meta-Analyses statement (Liberati et al., 2009; Moher et al., 2009). A study protocol was not published.

Search strategy

A search was performed using the following databases: Medline (via Ovid), EMBASE, PsycINFO, Psyndex, and the Cochrane Library. The following search terms were used: BZD(s) and/or z-drug(s) and/or anxiolyt*, hypnotic* in combination with information*, communicate*, educate*, support*, system*, aid*, program*, process*, material*, health intervent*, shared decision*, informed decision*, choice*, and train*. A sample syntax can be found in the appendix. The search was limited to studies published in English or German. The search began in September 2014 and was completed in October 2014.

Eligibility criteria

Studies were included in this review if they met the following criteria: had a controlled design, assessed middle-aged adults (45 years and older), used interventions focused on users of BZD or z-drugs and/or health care professionals (HCPs) involved in the care process, and had a primary outcome of interest of a reduction in BZD use and/or prescriptions. We excluded case series, review papers, meta-analyses, double publications, experimental research, protocols, and animal research. Moreover, studies were excluded if they focused on children or on chronically or seriously mentally ill patients, that is, if the use of BZDs was indicated (e.g., for severe psychiatric disorders such as schizophrenia). Psychopharmacological studies that examined medication phenomena only with respect to the drugs’ effects were also excluded. The types of interventions included were predominantly educational or informational in nature.

As part of our search strategy, we also performed a secondary search consisting of reference tracking for all full text documents included and a consultation of experts in the respective health care fields.

Study selection

First, duplicates were removed. Second, two independent researchers (AM, JT, or EC) screened the selected articles, first by title and then by abstract, for interventions related to the research topic. When the title and abstract were relevant or when eligibility was uncertain, the full text was retrieved. Any uncertainty concerning eligibility was resolved after an assessment of the full text and a discussion within the research team.

Data extraction and quality assessment

The collected data were extracted using a standardized sheet we had developed previously that was based on the Cochrane Extraction Form (Sambunjak Cumpston & Watts, 2017). The extraction form includes information about participants’ characteristics (age, gender), the treatment setting, inclusion and exclusion criteria, the randomization process, the intervention description, the duration of the intervention, outcomes, follow-ups, results, and significance. The interventions included were classified by the target population: BZD users, HCPs, or both groups. Data were extracted independently by two authors (AM and JT). Additionally, to consider the potential limitations of the studies included, the quality (or risk of bias) of these studies was assessed by two authors (AM and JT) using the Cochrane Collaboration’s tool for assessing the risk of bias in randomized trials (Higgins et al., 2011). The quality assessment form was based on six dimensions: random sequence generation, allocation concealment, blinding of participants, and personnel, blinding of outcome assessments, incomplete outcome data and selective reporting.

Data analysis

We used a qualitative analysis to synthesize the data extracted from the included studies (Dixon-Woods et al., 2005). Intervention approaches were classified into the following categories: those targeting patients, those with HCPs and multifaceted interventions. Furthermore, we subdivided the interventions into three patient-centered categories: physicians’ essential characteristics, clinician-patient communication, and patient information. A meta-analysis could not be conducted because the interventions were too heterogeneous.

Results

The review findings are presented in three steps. First, the studies are described and illustrated with charts. Then, they are subdivided into three sets, namely, patients, HCPs, and both groups combined. Next, the findings are described by an analysis of study quality, and then, the results are summarized in terms of patient-centered dimensions.

We identified 7,068 studies through the electronic search and 11 studies through our secondary search strategy. After the removal of duplicates (4,628) and after the screening process, 20 studies remained relevant and met the inclusion criteria (see Fig. 1).

Figure 1 Flow diagram of studies reviewed.

Description of identified studies

All studies were published in English between 1992 and 2014. The interventions were conducted in the UK (four studies), Australia (four studies), the USA (two studies), the Netherlands (two studies), Canada (two studies), Spain (three studies), Ireland (one study), Belgium (one study), and Sweden (one study). All studies were based on at least a controlled design. Eight studies used an explicit randomized controlled design, an additional nine used a controlled design (including intervention studies), and four used a cluster-randomized design. The study durations varied between 4 weeks and 29 months, with a mean of 6 months. Furthermore, the studies were conducted in different clinical settings that targeted inpatients, outpatients, community residents, or nursing home residents. The majority of the studies were conducted in general practices (11 studies) and nursing homes (five studies). One study each was carried out in a medical center, a hospital, an outpatient service (Medicaid), and a community pharmacy. While nine studies directly addressed BZD users (long-term, chronic, inappropriate), nine studies focused only on HCPs, specifically general practitioners and nurses. Two studies investigated the effect of interventions on both target patients and HCPs (physicians, nurses, and pharmacists). A systematic overview of relevant information for all interventions is shown in Tables 1–3.

Table 1 Description of included studies: patients.

Reference	Title	Location	Design	Setting	Duration (months)	Sample total n	Sample description: definition, mean age, sex distribution, groups	Intervention	Dimension of patient-centered-care model	Findings	
Bashir, King & Ashworth (1994)	Controlled evaluation of brief intervention by general practitioners to reduce chronic use of benzodiazepines	UK	CT	General practices	6	109	Chronic BZDs users, M = 62 years, 61% women intervention group (51) control group (58)	A self-help booklet included general information about benzodiazepine and techniques of coping with fears and anxiety supported with physician’s advice	Patient information	Eighteen percent of patients in the intervention group (9/50) had a reduction in benzodiazepine prescribing recorded in the notes compared with 5% of the 55 patients in the control group (p < 0.05)	
Cormack et al. (1994)	Evaluation of an easy, cost-effective strategy for cutting benzodiazepine use in general practice	UK	CT	General practices	6	209	Long-term regular users of BZDs, M = 69 years, 4:1 women to men letter group (65) letter plus advice group (75) control group (69)	Discontinuation letter asked the patient to reduce or stop the medication gradually and provide information about reducing medication and practical suggestions for nonpharmacological coping strategies plus 4-monthly information sheets	Patient information	After 6 months, both intervention groups had reduced their consumption to approximately two thirds of the original intake of benzodiazepines and there was a statistically significant difference between the groups. 18% of those receiving the interventions received no prescriptions at all during the 6 month monitoring period	
Gorgels et al. (2005)	Discontinuation of long-term benzodiazepine use by sending a letter to users in family practice: a prospective controlled intervention study	Nether-lands	CT	Family practices	6–21	4,416	Long-term BZDs users, M = 68 years, 65–69% women experimental group (2,595) control group (1,821)	Patient information as a discontinuation letter advised to gradually stop benzodiazepine use supported with patient-physician-communication, which evaluated actual benzodiazepine use	Patient information	At 6 months a large reduction in benzodiazepine prescription was present of 24% in the experimental group, vs. 5% in the control group. At 21 months again a steady reduction in benzodiazepine prescription of 26% was observed in the experimental group, vs. 9% in the control group, indicating that the short-term gain of the intervention was preserved	
Tannenbaum et al. (2014)	Reduction of inappropriate benzodiazepine prescriptions among older adults through direct patient education: The EMPOWER Cluster	Canada	RCT	Community pharmacies	6	303	Long-term BZDs users, M = 75 years, 69% women intervention group (138) control group (155)	Patient information via a personalized booklet comprising a self-assessment component including risks and advice about drug interactions and mentioning evidence, tapering recommendations and therapeutic substitutes as well as knowledge statements and peer champion theories to create cognitive dissonance about the safety of the benzodiazepine intake and augment self-efficacy	Patient information	At 6 months, 27% of the intervention group had discontinued benzodiazepine use compared with 5% of the control group (risk difference, 23% [95%CI, 14–32%]; intracluster correlation, 0.008; number needed to treat, 4). Dose reduction occurred in an additional 11% (95%CI, 6–16%)	
Ten Wolde et al. (2008)	Long-term effectiveness of computer-generated tailored patient education on benzodiazepines: a randomized controlled trial	Netherlands	RCT	General practices	12	695	Chronic BZDs users, M = 62.3 years, 68.1 women single tailored letter (163) multiple tailored letter (186) general practitioner letter (159)	Patient information either via two individual tailored letters aiming to reduce the positive outcome expectation of benzodiazepines by bearing in mind benefits of its withdrawal and in this case increasing self-efficacy expectations or a short general practitioner letter that modelled usual care	Patient information	Among participants with the intention to discontinue usage at baseline, both tailored interventions led to high percentages of those who actually discontinued usage (single tailored intervention 51.7%; multiple tailored intervention 35.6%; general practitioner letter 14.5%)	
Stewart et al. (2007)	General practitioners reduced benzodiazepine prescriptions in an intervention study: a multilevel application	Netherlands	CT	General practices	12	8,179	Chronic BZDs users, M = 64.63 years, 73.2% women intervention group (19 general practices) control group (128 general practices)	Patient information as a discontinuation letter outlined information about the risks of continuous use of benzodiazepines and recommended their withdrawal by inviting patients to an appointment to discuss this procedure, followed by an information leaflet about BZDs	Patient information and clinician-patient communication	Sending a letter to chronic long-term users of benzodiazepines advising decreasing or stopping benzodiazepine use in general practice resulted in a 16% reduction after 6 months and a 14% reduction after 1 year	
Heather et al. (2004)	Randomized controlled trial of two brief interventions against long-term benzodiazepine use: outcome of intervention	UK	RCT	General practices	6	284	Long-term BZDs users, M = 69.1 years, 48% females letter group (93) consultation group (98) control group (93)	Patient information via self-help booklet included information about tranquilizers, sleeping tablets and their withdrawal accompanied by a leaflet about sleeping problems and a discontinuation letter which informed about risks and advised to stop the intake, supported with patient-physician-communication including general information about benzodiazepines as well as advantages of and guidelines for withdrawal	Patient information and clinician-patient communication	Results showed significantly larger reductions in BZDS consumption in the letter (24% overall) and consultation (22%) groups than the control group (16%) but no significant difference between the two interventions	
Vicens et al. (2006)	Withdrawal from long-term benzodiazepine use: randomized trial in family practice	Spain	RCT	Public primary care centers	12	139	Long-term BZDs users, M = 59 years, 82% women intervention group (73)control group (66)	Patient information via physicians’ interview given on the first and follow up visits: first visit concentrated mostly on general information about benzodiazepines and their risks/effects, while the follow up visits focused on positive reinforcement of achievements	Patient information and clinician-patient communication	After 12 months, 33 (45.2%) patients in the intervention group and six (9.1%) in the control group had discontinued benzodiazepine use; relative risk = 4.97 (95% confidence interval [CI] = 2.2–11.1), absolute risk reduction = 0.36 (95% CI = 0.22–0.50). Sixteen (21.9%) subjects from the intervention group and 11 (16.7%) controls reduced their initial dose by more than 50%	
Vicens et al. (2014)	Comparative efficacy of two interventions to discontinue long-term benzodiazepine use: cluster randomized controlled trial in primary care	Spain	RCT	General practices	12	532	Long-term BZDs users, M = 64 years, 72% women structured intervention (SIF) (191) structured intervention with written instructions (SIW) (168) control group (173)	Educational intervention for patients with fortnightly follow-up visits to support gradual tapering (SIF) and written information material for patients rather than follow-up visits (SIW); patient information via educational interview included an information on benzodiazepine dependence, abstinence and withdrawal symptoms, risks of long-term use and reassurance about reducing medication as well as a self-help leaflet to improve sleep quality	Patient information, clinician-patient communication and essential characteristics of the clinician	At 12 months, 76 of 168 (45%) patients in the SIW group and 86 of 191 (45%) in the SIF group had discontinued benzodiazepine use compared with 26 of 173 (15%) in the control group. Both interventions led to significant reductions in long-term benzodiazepine use in patients without severe comorbidity	

Table 2 Description of included studies: health care professionals.

Reference	Title	Location	Design	Setting	Duration	Sample total n	Sample description: definition, mean age, sex distribution, groups	Intervention	Dimension of patient-centered-care model	Findings	
Avorn et al. (1992)	A randomized trial of a program to reduce the use of psychoactive drugs in nursing home	USA	RCT	Nursing home	5 months	823	Long-time users of psychoactive drugs and BZDs, not reported intervention group of 6 nursing homes (431) control group of 6 nursing homes (392)	Educational program to improve medical competence based on the principles of “academic detailing,” which focuses on direct patient care, alternatives to psychoactive drugs and recognition of adverse drug reactions face-to-face educational sessions by clinical pharmacists for prescribers and written information material for prescribers	Essential characteristics of the clinician and clinician-patient communication	Significant reduce psychoactive drug use in experimental group than in control (27% vs. 8%, p = 0.02). The comparable figures for the discontinuation of long-acting benzodiazepines were 20% vs. 9% (no significant)	
Batty et al. (2001)	Investigating intervention strategies to increase the appropriate use of benzodiazepines in elderly medical in-patients	UK	RCT	Hospitals	6–12 months	1,414	Inappropriate BZDs users, M = 75 years, not reported verbal intervention (not reported) bulletin intervention (not reported) control group (not reported)	Verbal intervention delivered in an interactive lecture format by a physician and a pharmacist to an audience arranged by the hospital contact. Bulletin intervention involved dissemination of printed material to physicians, pharmacist and nurses involved in the care at the hospital	Essential characteristics of the clinician and clinician-patient communication	Appropriate prescribing following verbal intervention increased substantially from 29% to 44% but this did not achieve statistical significance. There was a reduction in appropriate prescribing following bulletin intervention (42–33%) and no change following control intervention (42–42%)	
Berings, Blondeel & Habraken (1994)	The effect of industry-independent drug information on the prescribing of benzodiazepines in general practice	Belgium	RCT	General practices	4 weeks	128	General practitioners, not reported oral and written information (44) written information (43) no information (41)	Educational mail arguing for the rational and short-term prescribing of benzodiazepines, contained specific information regarding the limited effectiveness of long-term benzodiazepine use, risks and different forms of habituation and dependence supported by an independent medical representative whose oral message was congruent with the written materials and who answered any questions	Essential characteristics of the clinician	The absolute reduction in the number of prescribed packages was highest in condition one (oral and written information) with a mean decrease of 24% compared to the baseline. A reduction of 14% was found in physicians of condition two (written information) and of 3% in the control group	
Midlöv et al. (2006)	Effects of educational outreach visits on prescribing of benzodiazepines and antipsychotic drugs to elderly patients in primary health care in southern Sweden	Sweden	RCT	General practices	12 months	54	Physicians in general practices, not reported (not reported) intervention group (23) control group (31)	Physician’s and pharmacist’s visits in 2–8 week intervals: the first visit dealt with different causes of confusion in the elderly like medications, infections and other illnesses while discussing associated literature, whereas the second visit focused on the effects and risks of benzodiazepine use with medium or long acting duration of medication action	Essential characteristics of the clinician	One year after the educational outreach visits there were significant decreases in the active group compared to control group in the prescribing of medium- and long-acting BZDs and total BZDs but not so for antipsychotic drugs	
Pimlott et al. (2003)	Educating physicians to reduce benzodiazepine use by elderly patients: a randomized controlled trial	Canada	RCT	General practices	12 months	374	General practitioners, M = 50.6/50.7 years, not reported intervention group (168) control group (206)	Feedback packages were mailed that presented bar graphs comparing the prescriber with his or her peers and a hypothetical “best practice” supported by evidence-based educational material	Essential characteristics of the clinician	Although the proportion of long-acting benzodiazepine prescriptions decreased by 0.7% in the intervention group between the baseline period and the end of the intervention period (from 20.3%, or a mean of 29.5 prescriptions, to 19.6%, or a mean of 27.7 prescriptions) and increased by 1.1% in the control group (from 19.8%, or a mean of 26.4 prescriptions, to 20.9%, or a mean of 27.7 prescriptions) (p = 0.036), this difference was not clinically significant	
Pit et al. (2007)	A Quality Use of Medicines program for general practitioners and older people: a cluster randomized controlled trial	Australia	RCT	General practices	12 months	20 physicians	n = 20 general practitioners in 16 practices with n = 849 patients, older than 65 years intervention group (397) control group (352)	Educational sessions by pharmacists explaining how to conduct medication reviews with emphasis on benzodiazepines, accompanied by written sources of information on prescribing medication; risk assessment contained 31 items assessing risk factors for medication misadventure	Essential characteristics of the clinician	Compared with the control group, participants in the intervention group had increased odds of having an improved medication use composite score (odds ratio [OR], 1.86; 95% CI, 1.21–2.85) at 4-month follow-up but not at 12 months	
Roberts et al. (2001)	Outcomes of a randomized controlled trial of a clinical pharmacy intervention in 52 nursing homes	Australia	RCT	Nursing homes	12 months	52 nursing homes	52 nursing homes with n = 3.230 patients, not reported intervention group of 13 nursing homes (905) control group of 39 nursing homes (2 325)	Clinical pharmacy service model based on issues such as drug policy and specific resident problems, together with education and medication review and problem-based educational sessions for nurses addressing basic geriatric pharmacology and some common problems in long-term care medication; review by pharmacists highlighting the potential for adverse drug effects, ceasing one or more drug therapy, non-drug intervention and adverse effect and drug response monitoring	Essential characteristics of the clinician	This intervention resulted in a reduction in drug use with no change in morbidity indices or survival. The use of benzodiazepines was significantly reduced in the intervention group. Overall, drug use in the intervention group was reduced by 14.8% relative to the controls	
Smith et al. (1998)	A randomized controlled trial of a drug use review intervention for sedative hypnotic medications	USA	RCT	Medicaid recipients (outpatients)	6 months	189	BZDs users, 55 years and older, 61–63% women intervention group (99) control group (89)	Written information consisted of: a letter describing the drug use and education council guidelines for sedative hypnotic prescribing; a prescriber-specific profile about sedative hypnotic prescribing; a patient profile for each of the prescribers patients identified as over utilizers	Essential characteristics of the clinician	The intervention achieved a statistically significant decrease in targeted drug use, and the amount of reduction is likely to have decreased the risk of fractures associated with benzodiazepine use	
Smith & Tett (2010)	An intervention to improve benzodiazepine use—a new approach	Australia	CT	General practices (outpatients)	6 months	429 physicians	429 physicians intervention group (not reported) control group (not reported)	Information emails consisted of educational facts relating to benzodiazepines, including information on common side effects, indications, precautions and recommendations regarding prescribing as well as characteristics and alternative non-drug techniques; the website contained links to Australian Department of Health and Ageing websites which provided consumer information on medicines including sleeping tablets	Essential characteristics of the clinician	A significantly smaller number of aged care residents were on benzodiazepines for 6 months or more (p < 0.05) after the intervention compared with before	

Table 3 Description of included studies: patients and health care professionals.

Reference	Title	Location	Design	Setting	Duration (months)	Sample total n	Sample description: definition, mean age, sex distribution, groups	Intervention	Dimension of patient-centered-care model	Findings	
Patterson et al. (2010)	An Evaluation of an Adapted U.S. Model of Pharmaceutical Care to Improve Psychoactive Prescribing for Nursing Home Residents in Northern Ireland (Fleetwood Northern Ireland Study)	Ireland	RCT	Nursing homes	12	22 nursing homes	22 nursing homes with n = 334 residents, M = 82.7, 73% female intervention group (173) control group (161)	12 monthly visits from pharmacist to review prescription records of nursing home residents; collaboration of pharmacists with prescribers and patients to improve prescription patterns; pharmacist’s visits assessed the pharmaceutical care needs of each resident to identify potential and actual medication-related problems and reviewed the residents’ medication with the aim of optimizing psychoactive prescription	Essential characteristics of the clinician, clinician-patient-communication and patient information	The proportion of residents taking inappropriate psychoactive medications at 12 months in the intervention homes (25/128, 19.5%) was much lower than in the control homes (62/124, 50.0%) (odds ratio 50.26, 95% confidence interval 50.14–0.49) after adjustment for clustering within homes	
Westbury et al. (2010)	An effective approach to decrease antipsychotic and benzodiazepine use in nursing homes: the RedUSe project	Australia	CT	Nursing homes	6	25 nursing homes	25 nursing homes with n = 1,591 residents, not reported intervention group 13 nursing homes control group n = 12 nursing homes	Consciousness raising two drug use evaluation (DUE) cycles educational sessions promotional materials (newsletters, pamphlets, posters) and educational sessions and materials focused on informing health professionals and participants about risks and modest benefits associated with antipsychotic medications for dementia and benzodiazepines for sleep disturbance and anxiety management in elderly people	Essential characteristics of the clinician, clinician-patient communication and patient in-formation	Over the 6-month trial, there was a significant reduction in the percentage of intervention home residents regularly taking benzodiazepines (31.8–26.9%, p < 0.005). For residents taking benzodiazepines at baseline, there were significantly more dose reductions/cessations in intervention homes than in control homes (benzodiazepines: 39.6% vs. 17.6%, p < 0.0001)	

Quality assessments

The studies included in this survey differed considerably with respect to methodological quality (Higgins et al., 2011). Detailed evaluations for all studies are included in Table 4. Three categories were used to describe assessment quality: low, high, and unclear risk of bias (“yes” signified low risk; “no,” high risk; and “unclear,” all other cases). In a second step, quantitative levels were introduced; to meet the “low risk” level, all items in the question were required to have a low risk of bias. The “high risk” and “unclear” levels needed one item with a high risk of bias or an unclear risk of bias, respectively.

Table 4 Risk of bias.

Reference	RANDOM sequence generation	Allocation concealment	Blinding of participants and personnel	Blinding of outcome assessment	Incomplete outcome data	Selective reporting	
Interventions for patients	
Bashir, King & Ashworth (1994)	N.R.	H	H	U	L	U	
Cormack et al. (1994)	N.R.	L	H	U	U	U	
Gorgels et al. (2005)	N.R.	H	H	U	L	U	
Tannenbaum et al. (2014)	L	L	L	L	L	L	
Ten Wolde et al. (2008)	L	L	U	U	H	U	
Stewart et al. (2007)	N.R.	H	H	L	U	U	
Heather et al. (2004)	L	U	H	L	H	U	
Vicens et al. (2006)	L	L	H	U	L	U	
Vicens et al. (2014)	L	L	H	L	L	U	
Interventions for HCPs	
Avorn et al. (1992)	L	U	U	U	H	U	
Batty et al. (2001)	L	U	H	H	U	U	
Berings, Blondeel & Habraken (1994)	L	U	H	U	U	U	
Midlöv et al. (2006)	L	U	H	U	U	U	
Pimlott et al. (2003)	L	U	L	L	U	U	
Pit et al. (2007)	L	U	H	L	H	U	
Roberts et al. (2001)	L	U	H	U	H	U	
Smith et al. (1998)	L	U	H	U	H	U	
Smith & Tett (2010)	N.R.	U	H	H	H	U	
Interventions for patients and HCPs	
Patterson et al. (2010)	L	L	H	U	L	U	
Westbury et al. (2010)	N.R.	H	H	U	U	U	
Note:

Rating: L, low risk of bias; H, high risk of bias; U, unclear risk of bias; N.R., no relevance (controlled study design).

Regarding randomization, six studies were excluded from the assessment because of their study design (controlled trial) (Bashir, King & Ashworth, 1994; Cormack et al., 1994; Smith & Tett, 2010; Stewart et al., 2007; Westbury et al., 2010). In the remaining studies, the randomization was described clearly. Regarding allocation, six studies described in detail an allocation that was performed successfully (Cormack et al., 1994; Patterson et al., 2010; Tannenbaum et al., 2014; Ten Wolde et al., 2008; Vicens et al., 2006, 2014). Four studies reported an inappropriate allocation (Bashir, King & Ashworth, 1994; Gorgels et al., 2005; Stewart et al., 2007; Westbury et al., 2010). In the remaining studies, the allocation was unclear. Regarding the blinding of participants, only two studies performed this procedure adequately (Pimlott et al., 2003; Tannenbaum et al., 2014). Two other studies poorly described how the blinding process was carried out (Avorn et al., 1992; Ten Wolde et al., 2008). The remaining seventeen studies did not undertake any blinding of participants. Regarding the blinding of outcomes, six studies clearly blinded outcomes and documented the process well (Heather et al., 2004; Pimlott et al., 2003; Pit et al., 2007; Stewart et al., 2007; Tannenbaum et al., 2014; Vicens et al., 2014), two studies examined the outcomes in a nonblinded manner (Batty et al., 2001; Smith et al., 1998), and in the remaining 12 studies, it was unclear whether the respective outcomes had been blinded. The careful blinding in most studies may have impacted their results. Regarding incomplete outcome data, six studies were considered satisfactory, with a low probable risk of bias (Bashir, King & Ashworth, 1994; Gorgels et al., 2005; Tannenbaum et al., 2014; Vicens et al., 2006, 2014). In seven additional studies, outcome data were considered incomplete, increasing the risk of bias (Avorn et al., 1992; Heather et al., 2004; Pit et al., 2007; Roberts et al., 2001; Smith & Tett, 2010; Smith et al., 1998; Ten Wolde et al., 2008). Due to insufficient information, it could not be determined whether all patients in the remaining studies were included in the respective analyses; therefore, the risk of bias was unclear. Regarding selective reporting, only one study was found to have a low risk of bias (Tannenbaum et al., 2014). For the remaining studies, it was unclear whether important outcomes had not been produced or had simply not been reported.

In general, study quality was affected by a high risk of bias. Of the 29 studies in question, only one met all six categories to show no risk of bias (Tannenbaum et al., 2014). Seven studies were identified as having a low risk of bias in half of the categories, particularly those dealing with randomization and allocation and, to a lesser extent, the blinding of outcomes (Heather et al., 2004; Patterson et al., 2010; Pimlott et al., 2003; Roberts et al., 2001; Tannenbaum et al., 2014; Vicens et al., 2014). However, in these studies, the presentation of selective reporting was poor. The remaining 15 studies had a high risk of bias, mainly in the blinding of patients and personnel category. These studies also had poor presentations with respect to the blinding of outcomes and to incomplete data. Although most studies performed randomization well, a high risk of bias was prevalent in all five remaining categories. Thus, the overall quality of these studies, ranging from average to low, needs to be considered when interpreting their results. For the remaining valuation categories, all studies revealed vastly different standards of quality and poor presentation of procedures. If personnel and patients were not blinded, if the measurement processes became apparent, or if the results were not presented properly and completely, the effectiveness of the study in question could be compromised.

Summary of findings

The study results are presented again in terms of group subdivisions (patients, HCPs, both groups combined) and dimensions of patient-centered care. The data analysis identified three dimensions within the model of patient-centered care: patient information, clinician-patient communication, and essential characteristics of the clinician (Scholl et al., 2014).

Interventions concerning patients

Nine studies focused on patient interventions. Five studies examined the impact of patient information on the reduction of BZD use (Bashir, King & Ashworth, 1994; Cormack et al., 1994; Gorgels et al., 2005; Tannenbaum et al., 2014; Ten Wolde et al., 2008), while the remaining four studies looked at a combination of patient information and extra clinician-patient communication (Heather et al., 2004; Stewart et al., 2007; Vicens et al., 2006, 2014).

Patient information

Bashir, King & Ashworth (1994) demonstrated a short and simple intervention in which general advice from the GP combined with a self-help booklet reduced BZD intake after 6 months among patients who had taken the medication for more than a year. In a randomized controlled trial (RCT), Cormack et al. (1994) suggested that a letter containing information on BZDs and advice on how to reduce their intake, followed by 4 monthly information sheets, could reduce the intake of BZDs by approximately 1/3 after 6 months (Cormack et al., 1994). According to the authors, this simple method could significantly decrease intake among older people as well, whereas previous research suggested that such a reduction was harder to achieve. Another RCT with more than 4,000 participants showed that a letter with advice on how to gradually discontinue BZD use, followed by an appointment with the family practitioner to evaluate actual drug use, could significantly reduce participants’ BZD intake. A follow-up after 29 months confirmed the effectiveness of this intervention (Gorgels et al., 2005). In a subsequent RCT, Tannenbaum et al. (2014) suggested that a personalized eight-page patient-empowerment booklet, based on social constructivist learning and self-efficacy theory, supported the complete cessation of BZD use in older people. An overall reduction in BZD intake was observed 6 months after the intervention (Tannenbaum et al., 2014). Individually tailored interventions delivered to patients either once or three times in a row were effective at discontinuing BZD intake. Moreover, scientists from the Netherlands compared these tailored interventions to a short letter from a general practitioner and found that the former was superior (Ten Wolde et al., 2008).

Patient information and clinician-patient communication

Stewart et al. (2007) showed that a letter from a GP with a request to stop or reduce BZD use with their help coupled with a reminder 6 months later for those who had not responded significantly reduced the number of prescriptions per patient per half year. Nearly 150 practices and more than 8,000 patients were included in this study (Stewart et al., 2007). Heather et al. (2004) demonstrated how the dissemination of information to patients along with auxiliary educational talks with a GP could lead to a reduction in BZD intake within 12 months. BZD intake among older patients could be reduced in two ways: via patient information only or via patient information plus supportive communication from a physician. There was no significant difference between the first intervention with information on BZD provided by the GP (combined with a talk) and a second intervention consisting only of a letter signed by the GP. However, significant differences were found in a study that compared routine clinical practice to a treatment that contained standardized advice as well as a tapering-off schedule and biweekly follow-up visits (Vicens et al., 2006). At the 12 month follow-up, 45% of patients in the intervention group and 9.1% in the control group had discontinued their BZD use. This study concluded that the intervention was effective in terms of reducing long-term BZD use and was feasible in primary care. Vicens et al. (2014) conducted workshops that trained physicians how to interview patients and how to individualize patient information to lead to a gradual tapering of patients’ BZD intake. Regardless of whether patient consultations were followed by additional visits or written instructions, there was a reduction in long-term BZD use in patients without severe comorbidities (Vicens et al., 2014).

Interventions for health care professionals

Next, we systematically analyzed the studies that employed interventions aimed at HCPs and focused on their essential characteristics and clinician-patient communication as part of the patient-centered care model.

Essential characteristics of the clinician

Berings, Blondeel & Habraken (1994) conducted a study to assess whether oral and written information on BZDs or written information alone would have an effect on industry-independent information related to BZD prescribing among general practitioners. The statistical analysis suggested that the combination of physician contact and written information (24%) was superior to only written information (14%); both interventions together led to a decrease in the prescribing rate (Berings, Blondeel & Habraken, 1994). Midlöv et al. (2006) examined the effect of outreach visits. Experts visited physicians at private practices twice and provided them with information on confusion in older people and the effects of BZDs as well as other psychotropic drugs on this population (Midlöv et al., 2006). One year after the intervention, researchers found a significant decrease (25.8%) in the number of prescriptions of BZD. Pimlott et al. (2003) were interested in the effects of regular emails sent to physicians over a 6 month period with 2 month intervals. The email contained confidential profiles of BZD prescription users and educational bulletins (Pimlott et al., 2003). Physicians in the control group received educational bulletins related to antihypertension drug prescriptions for older people. The researchers reported a 0.7% decrease in prescribing rates in the intervention group and a 1.1% increase in the control group, but this difference was not significant. An educational program developed by Pit et al. (2007) evaluated an intervention complex that consisted of three major parts: educational resources (academic detailing, prescribing information, and feedback), medication risk assessments, and a medication review checklist (Pit et al., 2007). However, the intervention group did not show a significant reduction in the use of BZDs (OR = 0.51; 95%). Roberts et al. (2001) designed an approach to improve the quality of medication care among nursing home residents at large. This intervention consisted of three phases: the introduction to stakeholders of a new professional role related to relationship building, the education of nurses, and a medication review by pharmacists with a postgraduate diploma in clinical pharmacology. While the authors did not find a substantial change in morbidity indices or survival rates (primary outcomes), they did detect a significant decrease of 16.6% in BZD intake (14.8% in cumulative drug intake). Smith et al. (1998) investigated the effect of an intervention packet mailed to prescribers of BZDs. This package consisted of an intervention letter, a review of drug use, guidelines, and a prescriber-specific profile about the prescription of sedative hypnotics, as well as a patient profile for each of the prescriber’s patients who were identified as overutilizers. The researchers determined that this intervention significantly reduced the use of BZDs as a targeted sedative hypnotic medication in the intervention group (27.6%) versus a control group (8.5%). Smith & Tett (2010) investigated whether informing HCPs about BZD intake via emails and a website affected the number of BZD prescriptions over a 6-month period (Smith & Tett, 2010). After the intervention, there was a significantly smaller number of aged care residents who had used BZDs for 6 months or more (p < 0.05) but no significant change in the number of residents taking BZDs or taking BZDs for a long time and no significant change in the quantitative use of BZDs compared to the use among two different control areas (groups).

Essential characteristics of the clinician and clinician-patient communication

Avorn et al. (1992) found a significant reduction in the use of psychoactive drugs (BZD included) among residents at three nursing homes after they implemented a comprehensive educational outreach program (“academic detailing”) for HCPs. The reduction in BZD intake was 20% in the intervention group and 9% in the control group, and the patients in the intervention group reported reduced anxiety but more memory loss than the control group. Batty et al. (2001) investigated whether an interactive lecture or the dissemination of printed materials to physicians, nurses, and pharmacists would change the prescribing rate of BZDs toward a more appropriate rate for inpatients. Nearly 1,500 inpatients were included in the study. The prescribing rates were handled more appropriately in both intervention groups (intervention group 1: 29–44%; intervention group 2: 42–33%) than in a control group (42–42%), but these differences were not significant.

Interventions for patients and health care professionals

Finally, we identified two studies that employed a multifaceted approach toward both patients and HCPs that involved several dimensions of the patient-centered care model.

Essential characteristics of the clinician, clinician-patient communication, and patient information

Patterson et al. (2010) developed a multifaceted approach that entailed medication reviews by pharmacists over a 12-month period. The pharmacists’ visits consisted of a review of the residents’ prescribing information, the use of an algorithm to help prescribers assess the appropriateness of a medication, and individual conversations on improving prescriptions. As a result of the intervention, the proportion of residents taking inappropriate psychoactive medications at 12 months in the intervention group (25/128, 19.5%) was significantly lower (p < 0.001) than that in the control group (62/124, 50.0%) (odds ratio 50.26, 95% confidence interval 0.14–0.49) after adjustment for clustering within homes. No differences were observed at 12 months in the fall rate between the intervention group and the control group. Finally, these visits led to significantly lower rates of BZD prescribing and intake in the intervention group. In an RCT, Westbury et al. (2010) utilized a strategy from the Reducing Use of Sedatives project. This project involved a multistrategic interdisciplinary intervention for reducing the inappropriate use and promoting the appropriate use of medications that entails educational sessions, academic detailing, and a targeted sedative review. The intervention included raising awareness, two drug use evaluation cycles, educational sessions, promotional materials (newsletters, pamphlets, posters), academic detailing, and a targeted sedative review. This intervention complex led to a significant reduction in intervention home residents regularly taking BZDs (31.8–26.9%, p < 0.005) and antipsychotics (20.3–18.6%, p < 0.05); there were significantly more dose reductions and cessations in intervention homes than in control homes (BDZ: 39.6% vs. 17.6%, p < 0.0001; antipsychotics: 36.9% vs. 20.9%, p < 0.01) for residents taking BZDs and antipsychotics at baseline. In summary, the intervention of Westbury et al. (2010) led to a significantly higher rate of dosage reductions or cessations in intervention homes than in control homes.

Discussion

This review surveyed twenty interventions aimed at reducing the inappropriate prescription or use of BZDs and z-drugs. All interventions were based on patient-centered dimensions: patient information, clinician-patient communication, and essential characteristics of the clinician. We used the description of the interventions to assign them to the respective three dimensions of the patient-centered care model developed by Scholl et al. (2014). Patient-centered care is a broad concept in health care; this review shows that although there has been a growing focus on interventions that reduce the inappropriate use of BZDs and z-drugs, no study was defined as a patient-centered intervention or specifically measured the effects of such an intervention. Importantly, all included studies used a controlled design, and most showed a positive effect on the inappropriate prescription and use of BZDs and z-drugs for the intervention compared with typical care. There were comparisons between interventions and typical care as well as between interventions and other interventions. The interventions focused on patients showed a greater effect than those focused on HCPs. The studies that included both groups also showed a positive effect. This review suggests that patient-centered interventions that actively target patients, health professionals, or both are better than no intervention at all. Based on the results of this work, the following recommendations can be derived.

First, studies that examined patient information as one important dimension of patient-centered care and focused exclusively on patient-targeted interventions did not indicate a specific way to successfully reduce BZD and z-drug intake. In contrast, it has been shown that there are many methods to provide information that consider the patient’s informational needs and preferences. Studies have demonstrated that most educational interventions are more effective with middle-aged participants than with older participants (Mead & Bower, 2002; Meador et al., 1997). However, studies assessing elderly people show more diverse results than those without any age specifications (Mercer et al., 2008); therefore, there is a high probability that the effects of these interventions can also be achieved in older populations. The patient information studies established that providing patients (regardless of age) with information effectively led to the reduction or discontinuation of BZD and z-drug use, and this finding is consistent with previous research (Mugunthan, McGuire & Glasziou, 2011; Voshaar et al., 2006). Providing facts in a comprehensive and well-arranged way, as patient information does, encourages patients to consider reducing or discontinuing the use of the drug (Bodenheimer et al., 2002). Among the interventions that targeted patients, two studies supplemented the provision of patient information through consultations and active support by personnel; these studies also showed a significant reduction in BZD use. Providing patient information encourages patients to discuss these topics with their physician (Harter et al., 2011; Oshima Lee & Emanuel, 2013). Advising patients and discussing the best possible treatments are the main purposes of patient-centered care (Epstein, 2000; Scholl et al., 2014). The findings here emphasize the importance of providing patient information as part of a patient-centered approach (Farmer et al., 2008; Zill et al., 2015).

Second, the majority of the studies that focused on clinician-patient communication and essential characteristics of the clinician (HCPs) investigated interventions for HCPs; only three studies investigated interventions for patients. Studies that focused on patient interventions assessed a combination of patient information and clinician-patient communication and suggested that direct educational interventions and discussions with HCPs effectively reduces or stops inappropriate BZD use. This finding can be explained by the active participation of patients in the care process, as they are provided with all the information they need to make decisions regarding their medication consumption. Interventions targeting HCPs that include a combination of patient information sources (via e-mail, letter) and follow-up personal contact with HCPs provide models of success that may be more likely to be effective in reducing the inappropriate prescription and use of BZDs and z-drugs. This two-way communication is an important method of building practitioner-specific skills and increasing practitioner involvement in the interaction (Rao et al., 2007). Although, we did not explicitly describe and analyze secondary outcomes, in some of these combined studies, the most important results were the absence of symptoms (anxiety, distress, behavior disorders, life quality) as BZD usage was reduced (Avorn et al., 1992). The results were more varied with regard to interventions that concentrated on a set of verbal and nonverbal communication opportunities and skills and a set of attitudes, including those towards the patients, the HCPs themselves (self-reflection) and the medical competency of the HCPs. While some studies have found that the sole use of informative and educational training with printed educational material, training sessions and/or expert visits had positive effects on prescription rates and/or BZD use, other studies did not find similar results. However, it is possible that with educational efforts, positive changes with respect to the inappropriate prescription and consumption of BZDs can be achieved without disrupting care routines or producing high economic costs (Grimshaw et al., 2001). The factors associated with the knowledge and skills of prescribers belong to the most important dimension of patient-centered care. However, there are no conclusions concerning the comparison of effects between the significant studies. Most studies with statistically significant results used interventions that consisted of complex designs and methods, such as combinations of education and active individual exchanges about prescribing practices. These results suggest that an active exchange of knowledge during discussion leads to improvements in prescription habits. The duration of the studies that targeted clinician-patient communication and the specific characteristics of HCPs ranged from 5 to 12 months (one study lasted 4 weeks), suggesting that positive effects need time but will also be long-lasting. However, some of the studies that examined communication specifications or essential characteristics of HCPs did not report significant positive changes in prescription rates or the use of BZDs. A few explanations for these findings were provided (Batty et al., 2001; Pimlott et al., 2003), in particular, a focus on only one method of intervention (bulletin information) and a failure to combine several strategies. Furthermore, changes in prescribing habits associated with a long-term therapy (as with BZDs) are more difficult than in cases of acute and nonrecurring therapies, and some patients do not associate their medications with harmful effects. Therefore, more studies are needed that clearly define and describe the patient-centered dimensions of communication and HCP characteristics to allow for explicit comparisons and recommendations for clinical practice.

Third, this review included two multifaceted interventions that addressed patients as well as HCPs and examined three patient-centered dimensions of medical care: the essential characteristics of the clinician (HCP), clinician-patient communication and patient information (Patterson et al., 2010; Westbury et al., 2010). These studies demonstrated that inappropriate users who were actively informed about appropriate BZD use were more likely to reduce or discontinue BZD use. In addition, HCPs who were informed and involved in active exchanges improved their prescribing behavior, which is consistent with other reviews (Grimshaw et al., 2001). The available evidence indicates that interventions that address both patients and HCPs are effective and have significant positive effects if patient information and HCP education are implemented simultaneously (Joosten et al., 2008; Loh et al., 2007). The joint distribution of information and educational resources to both groups stimulates information exchange, which can lead to the cessation of drug use and/or improvements in prescribing behaviors (Cook et al., 2007; Stewart et al., 2000). Therefore, it is important to use a combination of strategies, such as updating HCP skills and improving awareness among patients, to help reduce or discontinue BZD and z-drug use. Other studies have found that interdisciplinary collaborations in medication-care-related interventions also improve drug use outcomes (Zwarenstein, Goldman & Reeves, 2009). However, these results should be interpreted with caution, as only two studies were included in the present analysis.

When analyzing the identified articles, it became clear that general practitioners and nursing homes were attempting to reduce the inappropriate use of BZDs and z-drugs. This finding was particularly true for older people who were being treated on an outpatient basis or by nursing home personnel.

As reported in other published reviews, a number of interventions capable of reducing BZD and z-drug use already exist (Mugunthan, McGuire & Glasziou, 2011; Voshaar et al., 2006). Interventions are more effective than routine care (Parr et al., 2006).

Consistent with previous reviews, interventions that target patients, which are represented under the dimension of patient information, have a positive effect on the reduction of BZD and z-drug use (Mugunthan, McGuire & Glasziou, 2011). A brief intervention in the form of either a letter or a single consultation is an effective strategy to decrease or stop inappropriate medication use without causing adverse consequences (Mugunthan, McGuire & Glasziou, 2011). Most strategies promote patient-centered care by providing information, boosting prescriber proficiency, and strengthening clinician-patient communication. Interventions that target patients and HCPs and use a multifaceted approach may be efficient, as studies of these interventions, in most cases, showed sustained reductions in BZD or z-drug use, consistent with other reviews (Gould et al., 2014). Our review emphasizes that there is a possibility of decreasing the inappropriate prescription and use of BZDs by providing patient-centered skills to providers. Finally, we found that effective interventions for changing clinical practice must target patients as well as HCPs and reflect the perspectives of patient-centered care (Dwamena et al., 2012; Legare et al., 2014).

Due to the heterogeneity of the included studies and their designs, this review did not attempt to compare the studies or make a final general statement. In addition, our findings and conclusions should be reconfirmed through further investigations.

Strengths and limitations

This is the first review of patient-centered care in the field of inappropriate BZD and z-drug usage. A systematic approach yielded a survey of patient-centered care interventions, providing a critical look at the multitude of methods that address different target groups along with their respective effectiveness. The quality of the studies suffered considerably from a lack of specificity. Study protocols were missing in all studies, and it was unclear whether all relevant information had been conveyed. Thus, it is necessary to be cautious when interpreting these results. This review focused on the primary outcome of a reduction in BZD and z-drug use and prescribing, and it did not consider secondary outcomes, such as the patients’ general health status (biological factors), social lives (social factors), or mental health status (psychological outcomes). The HCPs were also not analyzed in terms of their duration in the profession or their experience in treating older patients. An assessment of these factors is recommended in further scientific investigations to obtain a complete understanding of the problems involved in the inappropriate prescription and use of BZDs and z-drugs. Furthermore, one of the limitations is that although patient education seems to be more effective than approaches regarding HCPs, caution must be practiced with regard to generalization. A number of cognitively impaired older patients, especially in nursing homes (e.g., dementia patients), are not able to benefit from educational information. Finally, many studies were conducted using qualitative designs, and many were written in languages other than English; thus, these studies were not included in the current review, though they may also have been relevant. Therefore, future reviews should incorporate additional research designs.

Conclusion

The main finding of our systematic review is that patient information and educational strategies for HCPs can effectively lead to the appropriate use and prescription of BZDs. All three examined areas of patient-centered care (patient information, essential characteristics of the clinician, and clinician-patient-communication), alone or in combination, were generally effective at reducing and/or stopping the use of BZDs and z-drugs completely. These results suggest that inappropriate BZD and z-drug users (older adults) require and benefit from in-depth information about appropriate consumption. On the other hand, HCPs require more interventions in which they may communicate their clinical experiences with other groups of caregivers, discuss guidelines, and obtain additional knowledge to optimize their prescribing practices. Although this review focused on a patient-centered approach, it also revealed the limitations of studies that use this method. Before any final conclusions can be drawn, further investigations are needed to reconfirm the findings discussed here.

Supplemental Information

Supplemental Information 1 PRISMA checklist.

Click here for additional data file.

We would like to thank Eva Christale for her collaboration in the screening and data extraction processes.

Additional Information and Declarations

Competing Interests

Author Contributions

Data Availability

The authors declare that they have no competing interests.

Aliaksandra Mokhar conceived and designed the experiments, performed the experiments, analyzed the data, contributed reagents/materials/analysis tools, prepared figures and/or tables, approved the final draft.

Janine Topp conceived and designed the experiments, performed the experiments, analyzed the data, contributed reagents/materials/analysis tools, prepared figures and/or tables.

Martin Härter conceived and designed the experiments, authored or reviewed drafts of the paper.

Holger Schulz conceived and designed the experiments, supervision.

Silke Kuhn conceived and designed the experiments, authored or reviewed drafts of the paper, supervision.

Uwe Verthein conceived and designed the experiments, authored or reviewed drafts of the paper.

Jörg Dirmaier conceived and designed the experiments, performed the experiments, analyzed the data, contributed reagents/materials/analysis tools, authored or reviewed drafts of the paper, approved the final draft.

The following information was supplied regarding data availability:

This article is a systematic review and did not generate any data.

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
