# Peer review of "Patient-centered care interventions to reduce the inappropriate prescription and use of benzodiazepines and z-drugs: a systematic review"

_PeerJ, doi:10.7717/peerj.5535_

## Round 0.1 · original submission · Major Revisions

We apologize for the long delay it took us to reach a decision, but this was due to the necessity to obtain useful reports. As you will see, this eventually turned to your advantage as the reports submitted are both of high quality and constructive. Please, pay attention to all points raised and provide a detailed, point by point, rebuttal where you indicate exactly how your original submission has been modified as per the reviewers' comments.

Reviewer 1 ·

Basic reporting

This paper is a well structured and quality submission. The authors have taken a lot of time and effort preparing it. There are some aspects, however, that require clarification and addressing.
1. The non-English background of the authors is obvious with non-current terms used in some sections. E.g in abstract line 33 ‘side effect’ as opposed to ‘adverse effect’. In the introduction line 58 the word ‘tranquilizers’ is not in common use. May I suggest the use of the term ‘anxiolytic’ or ‘anxiolytic/hypnotics’ as an alternate. ‘Time horizon’, line 274, in the results section is also a term that is not in common use. In the discussion section, the word ‘inadequate’ line 333, should be replaced with ‘inappropriate’. The term ‘inadequate’ needs to be replaced by ‘inappropriate’ throughout the text. It’s used again in line 360 of the discussion. The use of the term ‘elderly’ is not current any more (line 378) and needs to be replaced with ‘older’.
2. Some of the background information about benzodiazepines and z-drugs is not correct and needs to be corrected. In the abstract it is stated (line 32) that long-term therapy is associated with considerable side effects. It needs to be stressed that short-term therapy is also associated with considerable side effects, including falls, language impairment, confusion, cognitive impairment and ataxia. The statement regarding risks being associated with long-term use only is also made in line 62 - intro. A statement regarding tolerance to the effects of benzodiazepines is needed in this section. There is also a line saying that z-drugs have similar pharmacological effects (Intro - line 61) to benzodiazepines. This is not true. Z-drugs have a different GABA receptor binding site and do not possess anxiolytic properties - so this statement is not correct.
3. The results section needs review as parts are not relevant, specific outcomes are not mentioned and some statements are unclear.
4. Table 2 is separately labelled ‘Table 1’ on its header page.
5. Finally, the referencing of journal names is inconsistent. Abbreviated versions are used alongside fully spelt non-abbreviated namings. Capital letters are also missing when Journal names are fully listed. A good edit for the correct style is needed.

Experimental design

The methodology of the review is a strength and well described in sufficient detail. The research question is well defined, relevant and meaningful. However,
1. In the methods section there is no inclusion of the search term ‘hypnot*’ (hypnotics) which is odd and may have resulted in some papers being missed, and
2. In the eligibility part of the methods line 121 it states that studies were excluded if they focused on BZD use for mentally ill patients (e.g. psychiatric diagnoses such as panic disorder or schizophrenia). This needs to be re-defined. Surely generalised anxiety disorder or sleep disturbance are psychiatric diagnoses as per the DSM-6 – In this case all studies would have to be excluded. Do the authors mean serious psychiatric disorders? Panic disorder would be a common diagnosis in older adults.

Validity of the findings

The validity is good and table data consistent with the studies. Interpretation is good. the conclusion links well to the introduction. Just a few points:
1. In the results section the authors confuse matters when they include antipsychotics in the discussion of the intervention results. E.g. Line 272 provides an example of this. “Meader et al investigated if an intervention could change the intake rate of antipsychotics…they found a reduction in antipsychotic dose (including BZD).“ It is not clear what the author means here and I question why the inclusion of antipsychotics in this results section is relevant to this paper?
2. When the interventions are discussed it would be clearer if the subject matter was confined to benzodiazepine use and the resulting outcome of the intervention. E.g. In the results section line 292, more detail is needed about the size of the decrease in drug use and whether the ‘drug use’ referred to was actually benzodiazepine use. In line 307 for this study there is no specific mention of benzodiazepines at all.
3. More detail and clarification is needed with line 310 which is the last sentence starting with 'Compared with a control group….'
4. Line 292, Smith is Alysia (i.e. female) so ‘his’ should be ‘her’. The information provided here in this discussion part stating that there was no effect of the intervention directly contradicts the information provided on Table 2 which reports a significant reduction in BZD use. Which outcome is it? Please make this clear.

Additional comments

A good proof reading by a native English speaker would assist this submission. The discussion section should also concentrate on the subject matter at hand and summarise the outcomes.

·

Basic reporting

This is a literature review of intervention studies, based on a patient-centred approach, and directed at patients and at Health Care Professionals (HCP) in order to reduce the chronic prescribing of Benzodiazepines and Z-drugs (by discontinuing chronic use and reducing incident BZD use).

Many features of a systematic review are present, but some are missing (protocol, common primary outcome measure, mathematical synthesis). Although this is not a Cochrane review, some essential features of Cochrane reviews have been used (trial selection procedure, risk of bias assessment of individual studies, Prisma flowchart and checklist).

The criteria of basis reporting are met, with regard to the language, refererences, background/context. The manuscript is well structured and self-contained.

Experimental design

Research question is well defined (for as far as patient-centred approach is well defined), and within the scope of the journal.

The investigation was rigorous and described in sufficient detail.

It is a bit more difficult to identify the knowledge gap that is identified here.

As the results are described only narratively, it is maybe not so important to analyse risk of bias across studies (publication bias, heterogeneity etc).

Validity of the findings

The narrative data are robust, but their statistical properties are hard to check.

The conclusions are within the fabric of the study and original research question.

The studies with inconclusive results are discussed. Replication and further research is encouraged.

The question of the novelty and impact of this work remains, but the overview of existing study within the framework of a theory (patient-centred approach) is useful.

The connection between theory and the simple and complex interventions is well discussed (although not with a theory of change map).

Additional comments

L 307 : positively influenced by some other variables" is not very informative.

332-334: what is the point here. That you only found studies in the setting of primary care (GPs) and nursing homes ? These sentences are unclear and wordy.
372 : do you mean "clinically relevant" or "statistically significant" ?

Some reference to reviews of pharmaco-epidomiological assessments of the risk of chronic BZD use and reviews of barriers and facilitators to use reduction could be made.

There is no distinction made between chronic not escalating low dosage sleeping pill use (only at bedtime) and escalating anxiolytic (or sedative) use in multiple doses per day.
These different indications should at least be discussed and also identified in the 21 selected articles.
Hypnotic* was not present in the search profiles.

---

## Round 0.2 · Minor Revisions

As you will see, one of the reviewers pointed out to several formatting and language problems. As PeerJ does not provide copyediting service, it is essential to fully correct that your paper before it can be published. I suggest that you contact one of your colleague who would be (preferably) English-native speaker or with a very good knowledge of English to polish your text. Also, use this opportunity to fine tune the paper as suggested by the reviewer. Lastly, make sure the references are shown correctly.

Reviewer 1 ·

Basic reporting

This is a better version.
-At times the English terms and expression are not correct e.g. derivates (should be derivatives) (69). The word 'seriously' needs to be put before 'mentally ill' (156). The word pharmacy should replace the word pharmacology (332). What do the authors mean by prescribing rates were handled more appropriately in both groups? (354) This statement doesn't make sense. The words 'monthly visits' (362) should be replaced by 'medication reviews'. The word 'greater' should replace 'clearer' (387). 'More profound' should be placed between to and improvements in line 436.
-Some statements are only relevant to the author country. e.g in the abstract it is stated that the maximum treatment is 8 weeks (41). In most English-speaking countries it is 4 weeks.
-Could the authors please include the results for combined interventions as they do with HCP and patient studies? What were the numerical impacts of the Patterson and Westbury studies specifically the relative reduction in benzodiazepine rates of use? This important information is omitted here for some reason?(359-373).
-citations are missing at times. e.g. behind the bold words in line 450, and after 'still prescribed frequently' (81)
- inconsistent referencing is still evident. Journal titles are not all capitalised and there is a big mismash of abbreviated titles and fully named journal titles. Please follow the referencing rules consistently
- Tables are sound.

Experimental design

Mostly well done review

Validity of the findings

Mostly good. I think though that the authors need to qualify that patient education, although generally a more effective strategy when compared to direct HCP approaches, may not be suitable for older cognitively impaired residents of nursing homes or people with advanced stages of dementia. This is not stated and I think it needs to be.

Additional comments

the PsycInfo database search engine should be written as PsycINFO - this is written several times in the text.

The authors have accepted most of the recommendations but it still needs some more editing work. They also really need to pay more attention to referencing. It makes the work look sloppy.

·

Basic reporting

no further comments, in addition to the first review round

Experimental design

no further comments, in addition to the first review round

Validity of the findings

no further comments, in addition to the first review round

Additional comments

The article has much improved. The authors have well responded to the comments of both reviewers.

---

## Round 0.3 · accepted · Accept

Your corrections have been made satisfactorily and, like the reviewer, I believe that your paper has been greatly improved. It will probably be an important piece of work to cite in the future. .

Reviewer 1 ·

Basic reporting

Thank you for working hard on the expression in English.

Experimental design

No comment

Validity of the findings

Revision made to conclusion.

Additional comments

Thank you for incorporating the comments of the reviewers so comprehensively. Your paper has benefited greatly from it. Thank you also for doing considerable justice to this important topic.